# Pb-Free Cs_3_Bi_2_I_9_ Perovskite as a Visible-Light-Active Photocatalyst for Organic Pollutant Degradation

**DOI:** 10.3390/nano10040763

**Published:** 2020-04-16

**Authors:** Bianca-Maria Bresolin, Carsten Günnemann, Detlef W. Bahnemann, Mika Sillanpää

**Affiliations:** 1Department of Separation Sience, School of Engineering Science, Lappeenranta University of Technology, Sammonkatu 12, 50130 Mikkeli, Finland; 2Institute of Technical Chemistry, Leibniz University Hannover, Callinstraße 3, D-30167 Hannover, Germany; guennemann@iftc.uni-hannover.de (C.G.); Bahnemann@iftc.uni-hannover.de (D.W.B.); 3Laboratory of Nano- and Quantum-Engineering (LNQE), Gottfried Wilhelm Leibniz University Hannover, Schneiderberg 39, D-30167 Hannover, Germany; 4Laboratory ‘‘Photoactive Nanocomposite Materials’’, Saint-Petersburg State University, Ulyanovskaya str. 1, Peterhof, 198504 Saint-Petersburg, Russia; 5Institute of Research and Development, Duy Tan University, De Nang 550000, Vietnam; mikaesillanpaa@gmail.com; 6Faculty of Environmental and Chemical Engineering, Duy Tan University, De Nang 550000, Vietnam; 7School of Civil Engineering and Surveying, Faculty of Health, Engineering and Science, University of Southern Queensland, West Street, Toowoomba, QLD 4350, Australia

**Keywords:** halide perovskite, laser photolysis, charge carrier separation, visible light activity

## Abstract

In our work, we employed Cs_3_Bi_2_I_9_ as a visible-light-active photocatalyst, synthesized with a low-temperature solvothermal method. The morphological and structural properties of the as-prepared perovskite were investigated, and the results were compared to previous studies to confirm its nature and the quality of the synthesis procedure. Transient absorption spectroscopy was applied in order to investigate the generation and lifetime of photogenerated charge carriers, revealing their formation after visible light excitation. The potential photocatalytic activity of the as-prepared metal halide perovskite was applied for the removal of Rhodamine B in aqueous solution, demonstrating an excellent activity of 93% after 180 min under visible-light irradiation. The current research aims to provide insights into the design of a new visible-light-active photocatalyst, Cs_3_Bi_2_I_9_, selected for its high application value in the field of advanced materials for light harvesting.

## 1. Introduction

The recent developments in industrialization have caused simultaneously remarkable energy and environmental challenges. This increasing global environmental pollution has become a pivotal issue; thus, sustainable technologies to solve the problem for human society should be developed [1]. On the other hand, the extensive employment of alternative renewable energy sources is also required for a sustainable future of our planet and its ecosystems.

Among the possible alternatives, heterogeneous photocatalysis has emerged as a promising technology holding the key for the future in environmental decontamination and renewable energy generation. One of its main advantages is the use of light energy, a sustainable source available all over the world [2,3,4,5]. Light energy is definitively considered one of the most promising sustainable sources, providing an environmentally friendly supply, and with its worldwide availability, it can be considered as one of the main candidates to overcome the future energy crisis [6,7].

In recent years, the use of light-activated photocatalysts emerged in heterogeneous advanced oxidation as great technology for environmental remediation and light energy conversion [8,9]. Some of the merits of these materials include the absence of fouling, the lack of mass transfer limitations, applicability at ambient conditions, and the ability to mineralize many organic pollutants into non-toxic compounds, such as water, carbon dioxide, and inorganic ions [10]. Among different photocatalysts, Titanium Oxide (TiO_2_) has attracted particular attention for many reasons, such as high stability, resistance to different pH conditions, non-toxicity, strong redox reaction potentiality, low cost, and availability in the market [2,11]. Unfortunately, with its wide bandgap (3.2 eV), conventional TiO_2_ could be mainly utilized under UV illumination that only accounts for a small fraction of the sun’s energy (5%) [12,13]. On the other hand, visible light accounts for almost 45% of the available solar radiation. Therefore, by exploiting novel visible-light-using active photocatalysts, sunlight would be more efficiently used, and the overall applications based on solar radiation would be greatly improved [14,15,16,17]. Thus, it seems profitable to design new and highly efficient visible-light-active photocatalysts for practical applications.

Recently, perovskite materials emerged for their record-breaking properties, in particular, their attractive light-harvesting ability [18,19,20]. Among them, lead halide perovskites have garnered a lot of attention for their remarkable performance, especially in photovoltaic devices. Unfortunately, their toxicity and instability remain two gaps to overcome for environmental applications [21,22,23]. In contrast, inorganic perovskite materials display much better stability [24,25,26]. Particularly, bismuth-based halide perovskites have gained attention by solving both the stability and toxicity issues related to lead-based materials [27,28,29]. Bismuth is known as an abundant metal on the earth’s crust; it can be recovered as a byproduct of other metals’ refining, and its price is quite stable and relatively low [30]. Several scientists have investigated ternary cesium bismuth halide (Cs_3_Bi_2_I_9_) as a new material for high-performance photovoltaic applications [27,31,32,33,34]. The Cs_3_Bi_2_I_9_ structure consists of identical perovskite-like fragments described by a general formula A_3_Bi_2_I_9_, with alternating edge-sharing [BiI_6_]^3−^ octahedral layers, where the voids are filled with Cs^+^ cations [35,36]. The tuneable structure and morphology have a great influence on the intrinsic electronic and optical properties that make this material so interesting in photocatalytic applications [37]. Many pioneers in photovoltaic technology have already experienced its advantages, such as excellent photoluminescence intensity suppression and significantly less toxicity than lead halides [31,33,34,37,38,39,40,41,42]. Intense research has been reported on Cs_3_Bi_2_I_9_ due to its attractive and impressive optical and electronic properties, such as excellent carrier transport behavior, high defect tolerance, and extremely low-density defects. However, several characteristics and potential applications of such an interesting material are still unrevealed.

The development of visible-light-active photocatalysts to tackle environmental contamination using a sustainable approach has been addressed as imperative, taking into consideration that visible light makes up the substantial fraction of the solar spectrum. In this work, we focused on Cs_3_Bi_2_I_9_ powder, prepared with an easily reproducible and low-temperature procedure. In particular, the visible-light-induced photogenerated charges were studied by laser flash photolysis revealing their mobility and dynamics effects.

Dyes are one of the main sources of water pollution [43,44,45,46,47]; in textile industries, around 12% of dyes used are found to be lost during operations every year, and 20% of their residues are estimated to be released directly in the environment [48]. High contamination has been reported for xanthene dyes, such as Rhodamine B (RhB), [49] which was found to be very harmful not only for the aquatic animals and plants, but also for humans, causing respiratory problems, asthma, dermatitis, mutagenicity, cancer, etc. [50]. Dyes show good solubility in water, and they are strongly resistant to the majority of conventional chemical and biological methods [51,52].

In our work, the as-synthesized metal halide perovskite was characterized by different techniques, and its photocatalytic activity was tested by studying the degradation of Rhodamine B in aqueous suspension under visible-light irradiation.

The aim of our work is, besides the characterization of the as-synthesized material, to investigate the ability of the perovskite to generate trapped charge carriers after excitation with visible light. Laser flash photolysis measurements were applied to determine the primary photochemical processes: the photogeneration of electrons and holes charge carriers. The reactivity of the trapped charge carriers was tested by the photodegradation of RhB.

The structure of the as-synthesized material is depicted in Figure 1. Based on our comprehensive study, we have proposed Cs_3_Bi_2_I_9_ as a potential visible-light photocatalyst justified by the investigated criteria, such as proper bandgap, suitable optical properties, and charge carriers’ dynamics.

## 2. Materials and Methods

### 2.1. Photocatalyst Preparation

Cs_3_Bi_2_I_9_ perovskite was synthesized from cesium iodide (CsI, 99.9% trace metals basis) and bismuth iodide (BiI_3_, purity 99%) using dimethylformamide (anhydrous, 99.8%, DMF) as the solvent. All chemicals were purchased from Sigma Aldrich (Darmstadt, Germany) and used without further purification. In a typical synthesis, CsI and BiI_3_ were mixed in dimethylformamide in a molar ratio of 3:2 and heated at 60 °C overnight for the evaporation of the solvent. Diethyl ether (purity ≥99.8%, Sigma Aldrich, Darmstadt, Germany) was used to wash the sample three times in order to remove DMF residues. Finally, the sample was dried in an oven at 60 °C overnight. The obtained material was finally ground in a mortar in order to obtain a fine powder.

### 2.2. Photocatalyst Characterization

The crystal phases and lattice parameters of the sample and its precursors were determined by powder X-ray diffraction using a D8 Advance diffractometer (Bruker, Solna, Sweden) provided with a Cu Kα radiation source. The UV–Vis diffuse reflectance spectrum of the as-synthesized perovskite was recorded with a Cary-100 Bio Spectrophotometer (Agilent, Santa Clara, CA, USA), equipped with an integrating sphere, at room temperature, within a wavelength range between 300 and 800 nm, and employing barium sulfate as the reflection standard. The optical bandgap energies were investigated using the Kubelka–Munk function [33]. The specific surface area of the sample was evaluated with Brunauer–Emmett–Teller (BET) measurements. The analysis was performed by a single point through adsorption of nitrogen at −196 °C, using a Chemisorb 2300 apparatus (Micro metrics Instrument). Information about the elemental composition of the powder and chemical state of the species on the surface of the sample was analyzed using X-ray photoelectron spectroscopy (XPS-Leybold Heraeus, Mellon, Toronto, ON, Canada) with 300 W AlKα radiation (20 eV pass energy with energy resolution 0.6 eV and 0.1 eV steps; modified Shirley background subtraction and modified Scofield relative sensitivity factors library were used in the quantification; data were processed with the advantage XPS analysis SW). Electrochemical measurements were performed in a three-electrode electrochemical cell provided with a Pt counter electrode and an Ag/AgCl/3 mol·L^−1^ NaCl reference electrode. Thin films of the Cs3Bi2I9 were prepared on top of fluoride-doped tin oxide (FTO)-coated glasses (Sigma Aldrich, ≈8 Ω sq^−1^) by the screen printing method [53]. The elemental composition was confirmed by means of energy dispersive spectroscopy (EDS). Transmission electron microscopy analysis was performed with a TECNAI FEI G2 microscope.

### 2.3. Laser Flash Photolysis Measurements

Transient absorption spectroscopy measurements (TAS) were performed with an Applied Photophysics LKS 80 Laser Flash Photolysis Spectrometer. In order to analyze the light absorption of the photogenerated transient species, the powder was excited with an Nd-YAG laser (Quantel; Brilliant B; 2rd harmonic, 532 nm) and analyzed with a pulsed xenon lamp (Osram XBO, 150 W; Austin, TX, USA). The diffusely reflected light was led to a monochromator and then to a photomultiplier as a detector (Hamamatsu PMT R928, Japan). Excitations energy densities of 3 mJ cm^−2^ per pulse were used and monitored by a Maestro energy meter (Gentec-EO). Resistance of 100 Ω was used for the connection between detector and oscilloscope. Before each experiment, the Cs_3_Bi_2_I_9_ powder sample was placed in a quartz flat cuvette flushed with nitrogen or methanol for 30 min according to the purpose of the measurements. The change in the reflectance ΔJ was obtained from the measured absorbance (Abs) values according to Equation (1), where I_0_ and I are the intensities of the reflected light before and after excitation, respectively.
(1)$$\Delta J\;=1-10^{-\mathrm{Abs}}=\frac{I-I_{0}}{I_{0}}$$

For each measurement, the average of 15 shots was considered, and the data points were reduced to 200.

### 2.4. Measurement of the Photocatalytic Reactivity

For the determination of the photocatalytic activity of the as-prepared sample, the photocatalytic conversion of Rhodamine B in aqueous solution was investigated at ambient temperature under visible-light irradiation. The experiments were performed in a 250 mL flask, with 100 mg of the photocatalyst powder that was dispersed into 100 mL of an RhB solution (20 mg/L). After a period to reach the adsorption equilibration in dark conditions (30 min), the surface of the solution was irradiated with a 150 W halogen lamp (Visilight CL150, Lutterworth, UK) with emission in the visible range (380–400 nm, Appendix A); more information is available in the Appendix A. The lamp was located at the top of the reactor at two centimeters from the surface of the solution. The decrease in the concentration of the target pollutant was monitored at constant intervals of time by a PerkinElmer Lambda (Waltham, MA, USA) 1050 UV—visible spectrophotometer.

## 3. Results and Discussion

### 3.1. Morphological Characterization

Figure 2 shows the recorded XRD pattern of the as-prepared material, which agrees with the reference pattern (01-070-0666 (I)—cesium bismuth iodide). Furthermore, no peaks corresponding to precursor materials were present (Appendix A), confirming the purity of the as-synthesized powder. The obtained lattice parameters are a = 8.404 Å, b = 8.404 Å, and c = 21.183 Å, which are in good agreement with previous studies [31,37]. More details are presented in our previous report [54].

The morphology of the Cs_3_Bi_2_I_9_ powder was investigated by means of transmission electron microscopy (TEM and SEM). Transmission electron microscopy images (Figure 3a,b) of Cs_3_Bi_2_I_9_ depict that the particles exhibited hexagonal shapes with an average size in the order of a hundred nanometers. From the SEM images, the Cs_3_Bi_2_I_9_ powder shows a typical 3D structure of particles with rough surfaces; further details are shown in our previous report [54]. The growth of well-defined crystals is important to highlight, because a high crystallinity, which is confirmed by the XRD as well, is one important factor for the efficient separation and transport of photogenerated charge carriers [55]. The charge recombination is affected by the presence of defects in crystals that behave as trap sites [56]. The present powder exhibits large but well-separated crystallized grains. These properties may facilitate the diffusion of free carriers and thus increase the efficiency of charge carriers under illumination [57]. Energy-dispersive X-ray spectroscopy (EDX) measurements were taken to verify the composition of the as-synthesized material (Figure 3c). According to the measurement, the element contents (I: 40.5%, Cs:12.4%, and Bi: 8.1%) were found to be in good agreement with the expected percentage [41].

The results of the XPS analysis of the as-synthesized Cs_3_Bi_2_I_9_ powder are shown in Figure 4, where the resulting overview survey is displayed. The spectrum shows mainly perovskite core peaks, but it includes some other contributions probably due to surface impurities and the contribution of the sample holder. As an example, the peaks around 293.5 and 287.2 eV can be associated with carbon or the two peaks around 550 eV are ascribable to oxygen. However, core-level spectra for Bi 4f, I 3d, and Cs 3d have been detected. The contribution of Bi 4f was detected in a range between 168.0 and 155.0 eV [58]. The smaller peak at the lower binding energy (157.2 eV) indicates the presence of a metallic Bi component, probably induced in part during the measurements [59]. Two intense peaks appear in the XPS spectrum related to the core level I 3d. According to previous studies, they can be associated with the doublets 3d_5/2_ and 3d_3/2_ with the lower binding component located at 618.9 eV and assignable to triiodide I_3_^−^. The additional broadening of the I 3d peak at 622.3 eV may indicate the presence of oxidized species of iodine at the surface, which forms the I_2_^+^ cation and the iodite anion (IO_2_^−^) [60]. The Cs 3d_5/2_ and Cs 3d_3/2_ peaks are observed at about 724.6 and 738.4 eV, respectively. The satellite broad peaks at higher binding energies of the main photoemission peaks, which may be identified as plasmons, arise from the excitation of plasma oscillations of the valence band electrons by the motion of photoelectrons in the solid. This feature was already shown in previous reports [61]. The XPS analysis further results in atomic percentages (I: 51.4%, Cs: 22.4%, and Bi: 9.9%) that agree well with the percentages obtained by the EDX analysis.

Further details are presented in our previous report [54].

The BET surface area of the Cs_3_Bi_2_I_9_ powder was measured based on nitrogen adsorption to be 5.6 m^2^·g^−1^.

As shown in the photograph (Figure 5), bare BiI_3_ displays a dark black color, whereas Cs_3_Bi_2_I_9_ is red. This evident color change strongly supports the formation of a BiI_3_ coordination complex, Cs_3_Bi_2_I_9_, when the precursors were dissolved in DMF [62,63].

### 3.2. Optical Characterization

UV–Vis diffuse reflectance spectra were recorded to quantify the light absorption ability and the optical bandgaps of the Cs_3_Bi_2_I_9_ powder; the results are shown in Figure 6a. The absorption starts at around 630 nm with an additional peak observed near 485 nm. Cs_3_Bi_2_I_9_ exhibits a strong excitonic behavior even at room temperature; the excitonic absorption peak at 2.56 eV (485 nm) may be related to the strong quantum confinement effect due to the 0D nature of the [Bi_2_I_9_]^3−^ bioctahedra [39]. The bandgap of the powder was calculated from the Tauc plot and was found to be indirect and equal to 1.95 eV (Figure 6b). Previous studies showed similar observations confirming the results herein obtained [28,31,57]. Moreover, a potential advantage is suggested in the use of Cs_3_Bi_2_I_9_ due to its ability to absorb light in the visible region. For example, TiO_2_, which is the most common photocatalyst, absorbs only in the UV region (λ < 400 nm) [64].

### 3.3. TAS

In order to get information about the processes occurring upon illumination, TAS measurements were performed in the microsecond time scale, which allows us to study the reaction dynamics of the photogenerated charges carriers. Because Cs_3_Bi_2_I_9_ was found to have a bandgap of 1.95 eV, the excitation was therefore performed with a 532 nm laser pulse. The measurements were performed in N_2_ atmosphere and in a mixture of N_2_ and methanol. In N_2_ atmosphere, the lifetime of the photogenerated carriers is maximized by preventing the reaction with donor or acceptor molecules, which are present in the air or adsorbed to the surface of the powder. When no electron donor or acceptor is present in the medium, the observed decay of the transient signals can be related to the internal recombination of charge carriers [65]. The addition of methanol, as a hole scavenger, makes it possible to identify at which wavelengths the trapped species (electrons and holes) absorb. The spectra in both atmospheres are shown in Figure 7a at 100 ns after the excitation. Both spectra have a similar shape and maximum, which can be observed at 620 nm in N_2_ atmosphere and in the presence of methanol at 640 nm. In general, the addition of methanol has a clearly visible effect on the ΔJ values, which are related to the number of photogenerated charge carriers, independent of the considered wavelength. In the range of 560 to 620 nm, the ΔJ values decrease after the addition of methanol, and thus, at these wavelengths, holes can be detected. Contrarily, between 640 and 740 nm, all ΔJ values increase in the presence of methanol, which is an indication for the presence of electrons at this wavelength range. The increase and the decrease of the ΔJ values for decay at a specific wavelength are not only detectable at a certain time point, but also present in the whole considered time scale, as can be seen for the decay at 640 nm (Figure 7b) and at 620 nm (Figure 7c), for example. It was reported for photocatalytic active materials such as TiO_2_ that 10 ns after the excitation, 90% of the electron–hole pairs were recombined and that the trapping of the charge carriers happened in the ps or fs time scale [66,67]. Because trapped charge carriers usually initiate photocatalytic redox reactions, the detection of trapped electrons and holes clearly supports the ability of Cs_3_Bi_2_I_9_ as a suitable material for photocatalytic reactions.

### 3.4. Photocatalytic Activity

To confirm the visible-light photocatalytic activity, which was proposed by the TAS measurements, the degradation of RhB as a model organic pollutant was studied. Figure 8a shows the change of the concentration of RhB as a function of the irradiation time for the blank experiment, the photocatalyst without light (dark), and the photocatalyst with visible light illumination (light). Without the presence of Cs_3_Bi_2_I_9_, only a negligible degradation of the dye can be observed in the considered time domain of 3 h. This is supported by the reported stability of RhB in aqueous solution under visible-light irradiation [68]. In the presence of the perovskite, but in the absence of light, only the adsorption of the dye molecules to the photocatalyst will take place. Within three hours, approximately 34% of the dye in the aqueous solution was adsorbed to the surface. The low amount of adsorbed dye molecules might be explained by the low surface area of the material. In contrast, in the presence of the perovskite and during visible-light illumination, a nearly complete degradation (93%) of the dye can be observed (Figure 8b).

For the application of a photocatalyst, recyclability is an important factor that needs to be considered as well. In Figure 9, the degradation of RhB is shown for three cycles, showing a degradation after 180 min of illumination of 93%, 92%, and 92% after the first, second, and third cycle, respectively. Just a small activity loss is present after three cycles with the perovskite material, which confirms the suitability of Cs_3_Bi_2_I_9_ for being used as a photocatalyst under visible-light illumination.

## 4. Conclusions

The present work describes the successful synthesis of Cs_3_Bi_2_I_9_ by a low-temperature and easily reproducible method, confirmed by an extensive morphological and optical characterization.

TAS measurements were applied in order to investigate the creation of photogenerated charge carriers, confirming that the material is not only absorbing visible light, but that the illumination also leads to the generation of electron–hole pairs, which can be trapped afterwards. The proposed activity was confirmed by an efficient degradation of RhB (93% after 180 min) during the illumination with visible light, confirming its potential as photocatalyst. After a recyclability test including three cycles, just a small activity loss was observed, which demonstrates the stability of the perovskite. All in all, Cs_3_Bi_2_I_9_ was shown to be a stable photocatalytic material under visible-light illumination, which is able to degrade RhB, making the perovskite an interesting material to investigate for the degradation of other compounds, as well. The aim of this study was to provide insights on a new photocatalyst by taking into account how environmentally friendly visible-light-responsive materials can offer the opportunity to revolutionize photocatalytic processes due to the major utilization of the potential solar spectrum.

## Figures and Tables

**Figure 1 nanomaterials-10-00763-f001:**
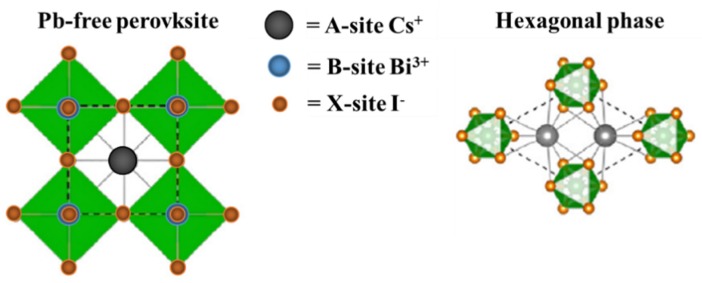
Illustrations depicting the crystal structures of the conventional perovskite and the hexagonal-phase-ordered-vacancy perovskite structure adopted by Cs_3_Bi_2_I_9_.

**Figure 2 nanomaterials-10-00763-f002:**
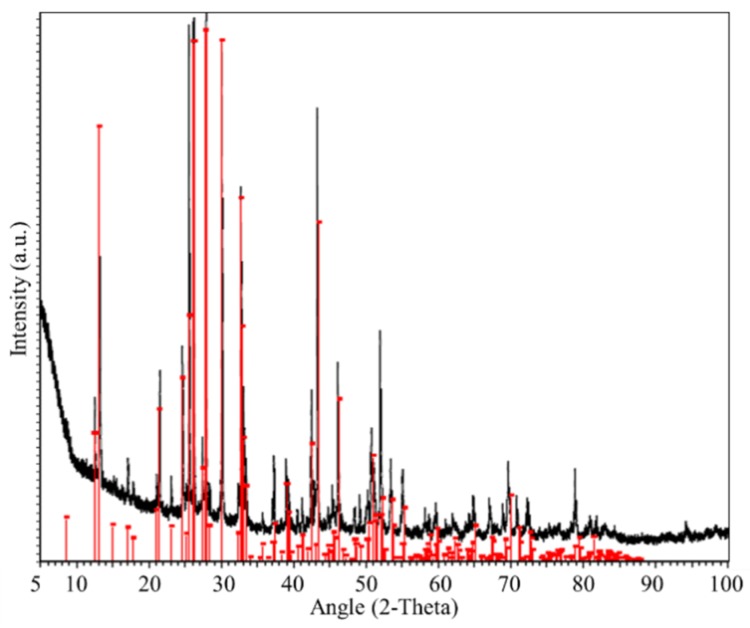
XRD pattern of Cs_3_Bi_2_I_9_ perovskite powder.

**Figure 3 nanomaterials-10-00763-f003:**
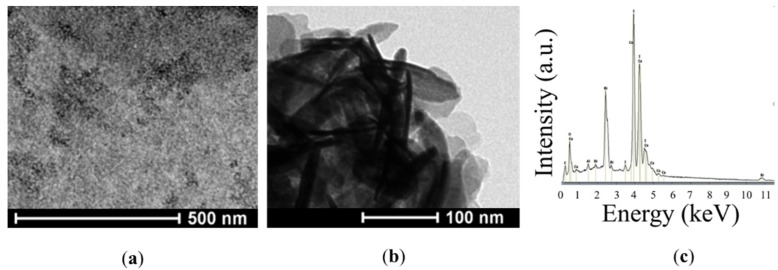
TEM images (**a**,**b**) and EDX analysis (**c**) of the as-synthesized Cs_3_Bi_2_I_9_ powder.

**Figure 4 nanomaterials-10-00763-f004:**
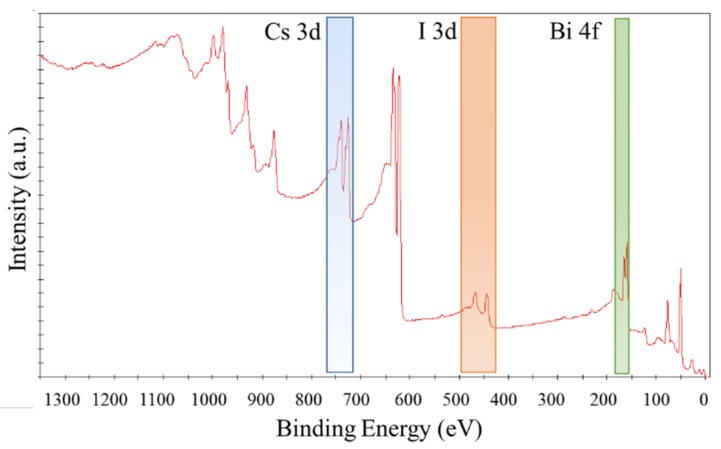
XPS survey spectrum.

**Figure 5 nanomaterials-10-00763-f005:**
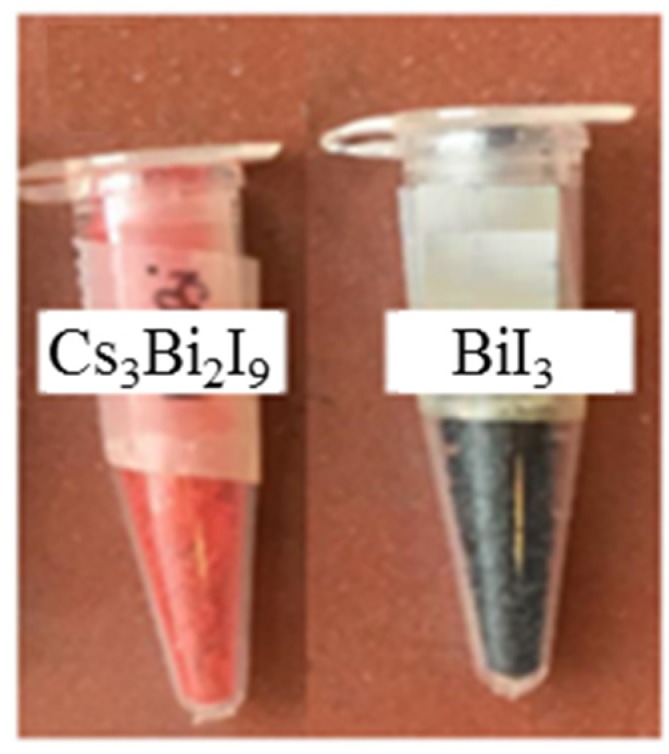
Dark bare BiI_3_ and red coordination complex Cs_3_Bi_2_I_9_.

**Figure 6 nanomaterials-10-00763-f006:**
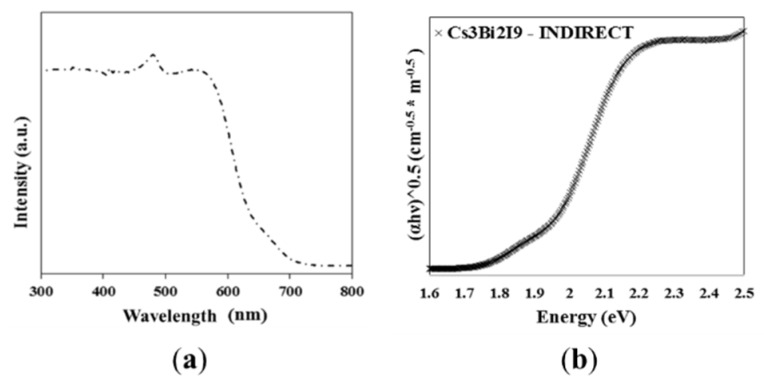
UV–Vis spectrum of Cs_3_Bi_2_I_9_ (**a**) and the Kubelka–Munk function for the determination of the bandgap energy of Cs_3_Bi_2_I_9_ (**b**).

**Figure 7 nanomaterials-10-00763-f007:**
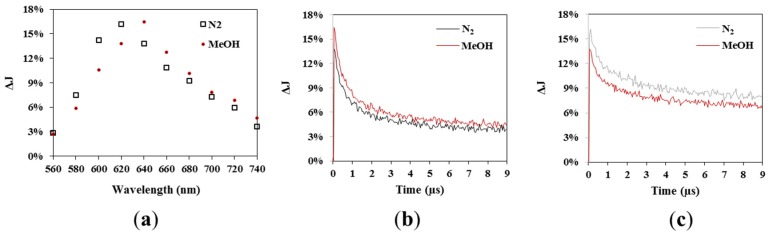
Transient absorption spectra in N_2_ and in methanol atmosphere at 0.1 µs after the excitation (**a**). Transient absorption signal measured in N_2_ and MeOH atmosphere at 640 nm (**b**) and at 620 nm (**c**).

**Figure 8 nanomaterials-10-00763-f008:**
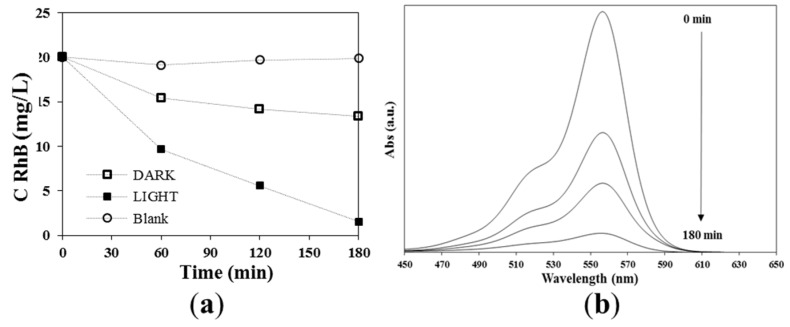
(**a**) Change in the concentration of Rhodamine B (RhB) at different conditions; (**b**) change in the absorption spectra of RhB during its photodegradation in the presence of Cs_3_Bi_2_I_9_ nanoparticles under visible-light irradiation.

**Figure 9 nanomaterials-10-00763-f009:**
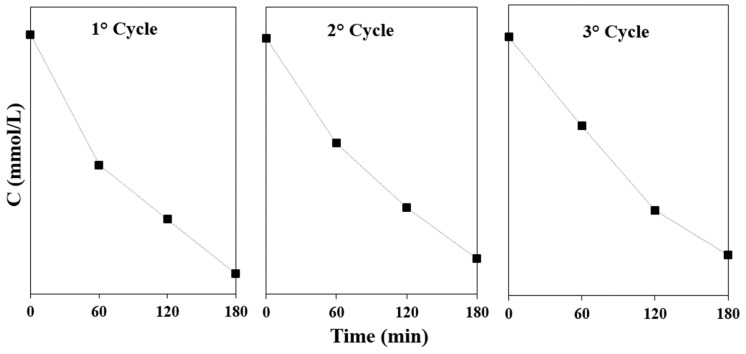
Recyclability test of the as-synthesized nanoparticles for RhB removal under visible-light irradiation.

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
