# Peer review of "Pb-Free Cs3Bi2I9 Perovskite as a Visible-Light-Active Photocatalyst for Organic Pollutant Degradation"

_nanomaterials, 2020, doi:10.3390/nano10040763_

Round 1

Reviewer 1 Report

This paper reports the synthesis, characterization and application of a Cs3Bi2I9 perovskite as photocatalyst for organic pollutant degradation in aqueous phase under visible light.

Although the development of new photocatalytic materials with enhanced visible light response is a hot topic nowadays, and this manuscript presents some results potentially interesting, it also presents important problems to be published in Nanomaterials journal.

*First of all, the work focuses on the use of the perovskite as visible light photocatalyst but in the introduction section it is highlighted the importance of the use of solar radiation. In this sense, the catalyst should be tested under solar radiation (or simulated) and compared to TiO2 to propose it as an effective catalyst in this application. The authors had a previous work (not cited in the manuscript, Solar Energy Materials & Solar Cells 204 (2020) 110214) in which bare Cs3Bi2I9 and also combined with TiO2 were tested under UV radiation, showing better results the combined catalysts. Is the application with only visible light so important if it is not completed with the study under solar radiation which is the final goal of the material?

*On the other hand, the synthesis of the perovskite was published and commented in the previous paper (which is more complet) even presenting some characterization results. Figure 3 is the same SEM and EDX from figure 3 of the previous paper of these authors in Solar Energy Materials & Solar Cells 204 (2020) 110214. Also XRD are reported in the previous paper. I understand this is not for self-plagiarism since in the previous paper this material was also combined with TiO2, but still needs to be removed from the manuscript and the previous work cited. The only novelty of the work is the application of the material for the removal of RhB under visible light radiation.

*In the introduction and experimental sections it is said that RhB and MB are used as model dyes, but only RhB results are presented. Only a few runs have been carried out and these results only partially support the conclusions about the reaction mechanism. In my oppinion they are speculative.

*Experimental details of the reaction set up are recommended (reactor dimenssions, stirring, irradiance, etc.) in order to compare the photocatalytic activity of the material with other systems.

*In the conclusions section, a whole paragraph about the reaction mechanism has been included but it seems that may be a different section if additional experiments are carried out to propose the mechanism.

*Even without taking into account the previous comments about the importance of testing the Cs3Bi2I9 perovskite under solar radiation, if the application of this material under visible light is proposed for the removal of dyes, additional experiments about the stability (leaching of organic carbon from the precursors, leaching of catalyst elements), mineralization of the organic compounds, more than 3-runs recycling, or even the effect of more realistic conditions are required.

Author Response

REVIEWER 1

This paper reports the synthesis, characterization and application of a Cs3Bi2I9 perovskite as photocatalyst for organic pollutant degradation in aqueous phase under visible light.

Although the development of new photocatalytic materials with enhanced visible light response is a hot topic nowadays, and this manuscript presents some results potentially interesting, it also presents important problems to be published in Nanomaterials journal.

* First of all, the work focuses on the use of the perovskite as visible light photocatalyst but in the introduction, section it is highlighted the importance of the use of solar radiation. In this sense, the catalyst should be tested under solar radiation (or simulated) and compared to TiO2 to propose it as an effective catalyst in this application. The authors had a previous work (not cited in the manuscript, Solar Energy Materials & Solar Cells 204 (2020) 110214) in which bare Cs3Bi2I9  and also combined with TiO2 were tested under UV radiation, showing better results the combined catalysts. Is the application with only visible light so important if it is not completed with the study under solar radiation which is the final goal of the material?

Thank you for your comments. Our research aim is to investigate a new material suitable for visible-light photocatalytic applications, since visible-light counts the major part of the solar spectrum and many reports have been published on materials able to absorb only in the UV-region. Furthermore, this research is meant to be a primary investigation on the potential of the chosen perovskite material by studying in particular the morphology, optical properties and the ability of generating trapped charge carriers. We have modified the introduction and the conclusion part to set the aim and focus of our study more clearly to the investigation of the visible light absorption properties.

* On the other hand, the synthesis of the perovskite was published and commented in the previous paper (which is more complete) even presenting some characterization results. Figure 3 is the same SEM and EDX from figure 3 of the previous paper of these authors in Solar Energy Materials & Solar Cells 204 (2020) 110214. Also, XRD are reported in the previous paper. I understand this is not for self-plagiarism since in the previous paper, this material was also combined with TiO2, but still needs to be removed from the manuscript and the previous work cited. The only novelty of the work is the application of the material for the removal of RhB under visible light radiation.

We thank you for your suggestions. We addressed your concerns in the manuscript. In particular, we modified the characterization part by providing TEM images instead of SEM images of the as prepared material and we have given a reference to the published paper to let the reader know that further characterization results have been reported there. This work is based on fundamental research of the possible ability of the perovskite for being used as visible light active photocatalyst. We want to show that this material is able to absorb visible light and that this absorption results in the formation of trapped charge carriers, which are able to degrade a model pollutant dye, as confirmation of the visible light activity.

* In the introduction and experimental sections it is said that RhB and MB are used as model dyes, but only RhB results are presented. Only a few runs have been carried out and these results only partially support the conclusions about the reaction mechanism. In my oppinion they are speculative.

Thank you for your comment. We have removed the parts in the introduction and the experimental section about methylene blue. We agree with your point that the results itself do not fully support the reaction mechanism and we have deleted that part from the paper. In general, the mechanism was included to give the readers of the paper an idea about the possible reaction steps, but after your concern we have decided that for future works the mechanism needs to be studied more in detail. The paper itself is not focussed on an application point of view, rather on fundamental research (see above).

* Experimental details of the reaction set up are recommended (reactor dimenssions, stirring, irradiance, etc.) in order to compare the photocatalytic activity of the material with other systems.

We thank you for your comments and we provided more information about the reaction setup in the manuscript at section 2.4 as recommended.

* In the conclusions section, a whole paragraph about the reaction mechanism has been included but it seems that may be a different section if additional experiments are carried out to propose the mechanism.

We agree with your point that for proposing a reaction mechanism more experiments are necessary (see above). This will be a part of future works and thus we decided to remove the part from the manuscript.

* Even without taking into account the previous comments about the importance of testing the Cs3Bi2I9 perovskite under solar radiation, if the application of this material under visible light is proposed for the removal of dyes, additional experiments about the stability (leaching of organic carbon from the precursors, leaching of catalyst elements), mineralization of the organic compounds, more than 3-runs recycling, or even the effect of more realistic conditions are required

Thank you for your comment. According to your point we have decided to set and explain the focus of our paper more clearly. As we have mentioned above the paper itself shall focus on fundamental research to provide information about the suitability of the perovskite for being used as photocatalyst. The photocatalytic experiments are used as confirmation to show that the detected visible light absorption and the formation of trapped charge carriers indeed results in a photocatalytic activity of the material.

Reviewer 2 Report

The manuscript «Pb-FREE Cs3Bi2I9 PEROVSKITE AS VISIBLE-LIGHT ACTIVE PHOTOCATALYST FOR ORGANIC POLLUTANT DEGRADATION» by B.-M.Bresolin, C.Günnemann, D.W. Bahnemann and M.Sillanpää is devoted to the synthesis and investigation of the photocatalytic activity of Cs3Bi2I9 perovskite. This is interesting work, which has exceptional importance for the development of catalysts for photocatalytic degradation of pollutants in water. The manuscript contains the experimental data obtained by various physico-chemical methods, including optical characterization of the perovskite, SEM, EDX and XPS analyses etc.

However, the manuscript needs major revision. I can recommend this manuscript for publication in Nanomaterials only after clarification and correction of the following points.

  1. Please correct A LOT (!!!) of grammatic misprints (see below)!

page 1, line 16, change “structuralpropertiesof” to “structural properties of”,

page 2, line 35 and page 16, line 295, change “contaminats” to “contaminants”,

page 2, line 43, change “stongly” to “strongly”,

page 3, line 62, change “exploting” to “exploiting” or “exploring” (?),

page 4, line 92, change “: “Illllustrations” to “Illustrations”,

page 5, line 106, change “precuresors” to “precursors”,

page 7, line 153 and page 8, line 162, please correct the font style,

page 8, line 171, change “This” to “These”,

page 9, line 180, change “spectum” to “spectrum”,

page 9, line 187, change “probabily” to “probably”,

page 9, line 192, change “oxidated spiecies” to “oxidized species”,

page 11, line 221, change “probabily” to “probably”,

page 12, line 240, change “Contrarlity” to “Contrarily”,

page 14, line 275 and page 16, line 306, change “acitivty” to “activity”,

page 15, line 294, change “processe” to “processes”,

  1. Please correct the numbering of the Figures

page 10, line 204, figure 5 is missed, figure 6 should be after figure 5,

page 12, instead of figure 6 should be figure 7,

  1. Please correct the numbering of the Sections

page 13, instead of section 3.2 should be section 3.4.

  1. Please correct the References

page 21, ref 34, don't start the ref. with "and",

page 24, ref 53, change “a. I. Zaitsev” to “A. I. Zaitsev”,

page 24, ref 50 and ref 54 is the same.

  1. Page 14, line 274, please, provide the quantitative data of the activity decreasing (percentage, etc.)
  2. In Introduction please add some information about the electronic effects, which determine the unique optical properties of Cs-Bi-perovskite materials.
  3. In “Material and methods” section it is mentioned that the decomposition of Methylene Blue was also investigated. However, this is in contradiction with “Results and discussion” section as this process is not discussed. Please, clarify that.
  4. In subsection 2.2 “Photocatalyst characterization” (line 108) add the information about UV–Vis diffuse reflectance spectroscopy (optical range and resolution, whether integrating sphere was used, etc).
  5. In subsection 3.2 “Optical characterization” please provide the comparative data of the light absorption ability of Cs3Bi2I9 and common (most popular) photocatalysts.
  6. It is high desirable to mention in “Introduction” or in “Results and discussion” section that the described samples are nanomaterials and the nanoscale effects determine their unique properties.

Author Response

REVIEWER 2

The manuscript «Pb-FREE Cs3Bi2I9 PEROVSKITE AS VISIBLE-LIGHT ACTIVE PHOTOCATALYST FOR ORGANIC POLLUTANT DEGRADATION» by B.-M.Bresolin, C.Günnemann, D.W. Bahnemann and M.Sillanpää is devoted to the synthesis and investigation of the photocatalytic activity of Cs3Bi2I9 perovskite. This is interesting work, which has exceptional importance for the development of catalysts for photocatalytic degradation of pollutants in water. The manuscript contains the experimental data obtained by various physico-chemical methods, including optical characterization of the perovskite, SEM, EDX and XPS analyses etc.

However, the manuscript needs major revision. I can recommend this manuscript for publication in Nanomaterials only after clarification and correction of the following points.

1.Please correct grammatical misprints (see below)!

page 1, line 16, change “structuralpropertiesof” to “structural properties of”,

page 2, line 35 and page 16, line 295, change “contaminats” to “contaminants”,

page 2, line 43, change “stongly” to “strongly”,

page 3, line 62, change “exploting” to “exploiting” or “exploring” (?),

page 4, line 92, change “: “Illllustrations” to “Illustrations”,

page 5, line 106, change “precuresors” to “precursors”,

page 7, line 153 and page 8, line 162, please correct the font style,

page 8, line 171, change “This” to “These”,

page 9, line 180, change “spectum” to “spectrum”,

page 9, line 187, change “probabily” to “probably”,

page 9, line 192, change “oxidated spiecies” to “oxidized species”,

page 11, line 221, change “probabily” to “probably”,

page 12, line 240, change “Contrarlity” to “Contrarily”,

page 14, line 275 and page 16, line 306, change “acitivty” to “activity”,

page 15, line 294, change “processe” to “processes”

We kindly thank you for the carefully reading and the correcting of the grammar. The manuscript has been modified according to your suggestions.

2.Please correct the numbering of the Figures

page 10, line 204, figure 5 is missed, figure 6 should be after figure 5,

page 12, instead of figure 6 should be figure 7.

We thank your for your comments regarding the numbering of the figures. We have modified the numbering of the figures in the manuscript.

3.Please correct the numbering of the Sections

page 13, instead of section 3.2 should be section 3.4.

Thank you for your comment. We have changed the number of the section to the correct one.

4.Please correct the References

page 21, ref 34, don't start the ref. with "and",

page 24, ref 53, change “a. I. Zaitsev” to “A. I. Zaitsev”,

page 24, ref 50 and ref 54 is the same.

Thank you for your suggestions. All three concerns have been modified in the reference section.

5.Page 14, line 274, please, provide the quantitative data of the activity decreasing (percentage, etc.)

We agree with your comment that it is important to give numbers as percentage to give a better overview for the reader how the activity is changing at the three different cycles. We have modified the manuscript by adding the numbers (percentage) of the degradation of RhB after 180 minutes of each cycle.

6.In Introduction please add some information about the electronic effects, which determine the unique optical properties of Cs-Bi-perovskite materials.

Thank you for your suggestions. We have added the required information about the electronic effects to the introduction.

7.In “Material and methods” section it is mentioned that the decomposition of Methylene Blue was also investigated. However, this is in contradiction with “Results and discussion” section as this process is not discussed. Please, clarify that.

We agree with your point and have deleted the methylene blue parts from the experimental section and from the introduction (see also Reviewer 1).

8.In subsection 2.2 “Photocatalyst characterization” (line 108) add the information about UV–Vis diffuse reflectance spectroscopy (optical range and resolution, whether integrating sphere was used, etc).

We kindly thank you for your comment. We have provided more information concerning the UV vis measurement of the sample in the manuscript.

9.In subsection 3.2 “Optical characterization” please provide the comparative data of the light absorption ability of Cs3Bi2I9 and common (most popular) photocatalysts.

Thank you for your comment. We have added a part to our manuscript comparing our perovskite material with the most common photocatalyst TiO2.

10.It is high desirable to mention in “Introduction” or in “Results and discussion” section that the described samples are nanomaterials and the nanoscale effects determine their unique properties.

Thank you for your suggestion. We have added to both sections a statement about the size. See for example the added TEM images of the material.

Reviewer 3 Report

The present manuscript deals with a new application of a microsized perovskite material Cs3Bi2I9 as a photocatalyst for the removal of organic dye pollutants from water. The paper is well organized, the presentation is mostly clear and can be easily traced, and the conclusions drawn are supported by the shown material. There are only some minor points of criticism that theauthors should improve prior to a publication in this special edition of nanomaterials as follows:

1) Introduction, page 2, line 36: "annual production of 800.000" - a physical unit is missing, like "kg", "tons" or equivalent.

2) Page 5, photocatalyst characterization in section 2.2: For the XPS analysis, some more details on the experimental conditions and the data analysis are needed: What is the actual energy resolution of the instrument? What kind of background subtraction was used for quantification (i.e. linear, Shirley, Tougaard)? What are the photoionization cross sections used for quantification? Which software was used? Please add related information!

3) Page 6, section 2.3, line 128: Please include "... and the to a photomultiplier as detector".

4) Line 136/137: "... and the data points were reduced to 200." This is unclear to me - please specify.

5) Page 7, XRD analysis: How were the lattice parameters determined? Did the authors used a Rietveld refinement or an entire pattern fitting model? Which software was used, and what are the uncertainties of the determined lattice parameters - please add information.

6) Page 8, Fig. 2: Please adapt the scaling of the patterns from the three diffractograms to the same height, the pattern of the most interesting actual sample is the smallest ...

7) Page 8, line 174-176: What are the uncertainties for the quantification? Please specify an error estimate.

8) Page 9, Fig. 3: The scaling and the labelling of the EDX are far too small, please enlarge labels.

9) Page 9, discussion of XPS peak energies: Please be more precise in the binding energies, at least it should read e.g. "... the Bi 4f peaks at 160.2 eV and 165.5 eV." (line 188), at least this is what the figure suggests to me.

10) Page 9, line 194: Remove "photoemissive" and insert "photoemission"

11) Page 10, line 196-198: What are the errors of the quantitative XPS analysis? I cannot believe those values, as the background lines in Fig. 4 are highly erroneous in particular for the I 3d but especially for the Cs 3d lines, the background cuts substantial parts of the peaks. Furthermore, are the entire peak groups used for the quantification, i.e. both peaks for Bi3+ and Bi metal?

12) Page 10, Fig. 4(a), please label the detected and assigned peaks. Please also justify the use of three peaks for the deconvolution of the I 3d. Where the plasmon peaks for Cs included in the quantification? The Figure caption is erroneous as (C) and (D) are mixed up.

13) Page 15/16, the numbering of the equations is wrong, eq. 1 was already used on page 6, and thus it should read eq. 2, 3 and 4 on the latter pages.

14) There are several typos and misspellings:

page 1, line 16 "structuralpropertiesof"  -- please use three words!

page 2, line 35 "Contaminants"

page 2, line 38 "were" instead of "was"

page 2, line 43 "strongly" (the "g" is missing)

page 3, line 73 it should read "recovered"

page 3 last line, page 4, first line: it should read " ..., its alternating ..."?

page 7, line 155: please remove one point (..)

page 8, line 171, it should read "crystallized"

page 8, line 172, it should read "... and thus increase ..."

page 9, line 181, it should read "sample holder" instead of just "holder"

page 9, line 187, it should read "probably"

page 9, line 192, it should read "species"

page 12, line 240, it should read "Contrarily"

page 12, line 241, it should read "...which is an indication for the presence of electrons ..."

page 13, line 247, it should read "usually"

page 14, line 267, it should read "in the presence of"

page 15, line 286, it should read "three processes"

Provided that the authors seriously consider the comments given above and provide a revised mansucript accordingly, I think that their work can be a nice contribution to "nanomaterials" - although their particles are in fact not in the "nano-" but in the "microscale".

Author Response

REVIEWER 3

The Authors examined the possibility of using Pb-free Cs3Bi2I9 perovskite as visible-light active photocatalyst for Rhodamine B degradation in aqueous solution. The research presented deals with an up-to-date environmental issue, and in my opinion reveals a medium level of novelty. However, I believe that the following major points should be considered:

  • Aim, novelty, and impact of such study are not clearly highlighted.

Thank you for your comment. As mentioned based on the comments of Reviewer 1, our aim is to provide a fundamental study of the chosen perovskite material. We want to show that the material is able to absorb visible light and that this results into the formation of photogenerated and trapped charge carriers, which are able to drive photocatalytic reactions, as shown in this case by the degradation of RhB. We have modified the manuscript to set the aim and the impact of the study more clearly.

  • No data on the lamp emission (specific wavelength range) and the lamp power (Einstein/time) are provided. It is therefore not possible to evaluate the photoefficiency under visible light of the process and compare the results with other literature findings.

Thank you for your suggestions. We agree with your point and have added more information about the lamp to the Supporting Information.

  • Poor information (in the Conclusions) on reaction mechanisms have been hypothesized and described. This point should be deepened within a Discussion section.

Thank you for your comment. Based on the comments of Reviewer 1 we have decided to not include the mechanism. We agree with the fact that a mechanism might be important for the degradation of organic pollutants. In general, the focus of our work is to use the photocatalytic experiments as confirmation of the proposed activity based on TAS. For clearly proposing the mechanism further experiments are necessary, which will be a part of future works.

  • No characterization analyses have been reported on the catalysts after their use. They are crucial to prove with scientific certainty photocatalyst stability and recyclability in view of industrial applications.So can we let the answer you gave. I would really avoid to put more comments on the size if not required specifically

Thank you for your comment. Our research aims to the investigation of the suitability of the perovskite as visible light active photocatalyst. We want to show that indeed the material is able to create trapped charge carriers upon visible light absorption, which are able to drive chemical reactions. The degradation of RhB was used as a test experiment to confirm the former results. Thus the manuscript is dealing more with fundamental research. We have modified the manuscript for making the scope more clear.

  • In its current state, the level of English throughout the manuscript should be improved. Several typos should be corrected.

Thank you for the suggestions, we have modified the manuscript to increase the level of English and we have corrected typos.

Reviewer 4 Report

The Authors examined the possibility of using Pb-free Cs3Bi2I9 perovskite as visible-light active photocatalyst for Rhodamine B degradation in aqueous solution. The research presented deals with an up-to-date environmental issue, and in my opinion reveals a medium level of novelty. However, I believe that the following major points should be considered:

  • Aim, novelty, and impact of such study are not clearly highlighted.
  • No data on the lamp emission (specific wavelength range) and the lamp power (Einstein/time) are provided. It is therefore not possible to evaluate the photoefficiency under visible light of the process and compare the results with other literature findings.
  • Poor information (in the Conclusions) on reaction mechanisms have been hypothesized and described. This point should be deepened within a Discussion section.
  • No characterization analyses have been reported on the catalysts after their use. They are crucial to prove with scientific certainty photocatalyst stability and recyclability in view of industrial applications.
  • In its current state, the level of English throughout the manuscript should be improved. Several typos should be corrected.

Author Response

REVIEWER 4

The present manuscript deals with a new application of a microsized perovskite material Cs3Bi2I9 as a photocatalyst for the removal of organic dye pollutants from water. The paper is well organized, the presentation is mostly clear and can be easily traced, and the conclusions drawn are supported by the shown material. There are only some minor points of criticism that theauthors should improve prior to a publication in this special edition of nanomaterials as follows:

1) Introduction, page 2, line 36: "annual production of 800.000" - a physical unit is missing, like "kg", "tons" or equivalent.

Thank you for your suggestion, as suggested also from the other reviewers we have modified this section of the manuscript to make it more consistent with the scope of this study.

2) Page 5, photocatalyst characterization in section 2.2: For the XPS analysis, some more details on the experimental conditions and the data analysis are needed: What is the actual energy resolution of the instrument? What kind of background subtraction was used for quantification (i.e. linear, Shirley, Tougaard)? What are the photoionization cross sections used for quantification? Which software was used? Please add related information!

Thank you for the comment. We have add information in the manuscript and modify the XPS measurement presentation.

3) Page 6, section 2.3, line 128: Please include "... and the to a photomultiplier as detector".

Thank you for your comment. We have included your suggestion to the experimental section.

4) Line 136/137: "... and the data points were reduced to 200." This is unclear to me - please specify.

Thank you for your comment. This sentence means that the number of saved data points is reduced to 200 based on the used software. In general, this is an automatically step, where raw data points are reduced to less points in the saved file.

5) Page 7, XRD analysis: How were the lattice parameters determined? Did the authors used a Rietveld refinement or an entire pattern fitting model? Which software was used, and what are the uncertainties of the determined lattice parameters - please add information.

Thank you for your comment. The baseline was adjusted and the position of some peaks. No refinement has been done in the patterns and the software used to present the results is Diffrac.EVA. About the reference, we can supply the following data: Calculated from ICSD using POWD-12++ (2004), Chabot, B., Parthe, E., Acta Crystallogr., Sec. B, volume 34, page 645 (1978).

6) Page 8, Fig. 2: Please adapt the scaling of the patterns from the three diffractograms to the same height, the pattern of the most interesting actual sample is the smallest.

Thank you for the suggestions, we modified the presentation of the data to make it more clearly for readers. For further information, we are referring to the previous work in the manuscript.

7) Page 8, line 174-176: What are the uncertainties for the quantification? Please specify an error estimate.

Thank you for the suggestions, we have added the information to the manuscript.

8) Page 9, Fig. 3: The scaling and the labelling of the EDX are far too small, please enlarge labels.

Thank you for your comment. We agree with your point and we have modified the figure.

10) Page 9, line 194: Remove "photoemissive" and insert "photoemission"

Thank you for your suggestion. We have changed the word in the manuscript.

9) Page 9, discussion of XPS peak energies: Please be more precise in the binding energies, at least it should read e.g. "... the Bi 4f peaks at 160.2 eV and 165.5 eV." (line 188), at least this is what the figure suggests to me.

11) Page 10, line 196-198: What are the errors of the quantitative XPS analysis? I cannot believe those values, as the background lines in Fig. 4 are highly erroneous in particular for the I 3d but especially for the Cs 3d lines, the background cuts substantial parts of the peaks. Furthermore, are the entire peak groups used for the quantification, i.e. both peaks for Bi3+ and Bi metal?

12) Page 10, Fig. 4(a), please label the detected and assigned peaks. Please also justify the use of three peaks for the deconvolution of the I 3d. Where the plasmon peaks for Cs included in the quantification? The Figure caption is erroneous as (C) and (D) are mixed up.

Thank you very much for the careful attention on the details of the data. We simplified the presentation of the mentioned results. We referred to previous work for further characterization. We kindly thank you for the efforts you have made to explain your comments.

13) Page 15/16, the numbering of the equations is wrong, eq. 1 was already used on page 6, and thus it should read eq. 2, 3 and 4 on the latter pages.

Thank you for your suggestion. Based on the comments of the others reviewers we have decided to remove the mechanism part form the paper, since more experiments for clearly describing the steps are necessary, which will be a part of future works.

14) There are several typos and misspellings:

page 1, line 16 "structuralpropertiesof"  -- please use three words!

page 2, line 35 "Contaminants"

page 2, line 38 "were" instead of "was"

page 2, line 43 "strongly" (the "g" is missing)

page 3, line 73 it should read "recovered"

page 3 last line, page 4, first line: it should read " ..., its alternating ..."?

page 7, line 155: please remove one point (..)

page 8, line 171, it should read "crystallized"

page 8, line 172, it should read "... and thus increase ..."

page 9, line 181, it should read "sample holder" instead of just "holder"

page 9, line 187, it should read "probably"

page 9, line 192, it should read "species"

page 12, line 240, it should read "Contrarily"

page 12, line 241, it should read "...which is an indication for the presence of electrons ..."

page 13, line 247, it should read "usually"

page 14, line 267, it should read "in the presence of"

page 15, line 286, it should read "three processes"

Thank you very much for the careful suggestions given. We have modified all concerns in the manuscript.

Provided that the authors seriously consider the comments given above and provide a revised mansucript accordingly, I think that their work can be a nice contribution to "nanomaterials" - although their particles are in fact not in the "nano-" but in the "microscale"

Round 2

Reviewer 1 Report

The manuscript has been improved according to the considerations done. I still think that there is little effort in the photocatalytic activity section, but, as the authors state this can be considered a preliminary study for future applications. Therefore, my new consideration is that the paper can be accepted for publication in Nanomaterials journal.

Author Response

Please revised the manuscript according to the Reviewers indications:

Underline the aim, novelty, and impact the study in the introduction section.

Thank you for your comment. We highlighted parts of the introduction and the abstract in different colours to mark the aim, novelty, and the impact.

Specify the data about the lamp emission ( wavelength range) and the lamp power (Einstein/time)

Thank you for your suggestion. We added the data to section 2.4 (lines 156-157).

Moreover, several typos should be corrected.

Thank you for your suggestion. We corrected the typos to improve the quality.

Other minor revision are reported below:

1) Introduction, page 2, line 36: "annual production of 800.000" - a physical unit is missing, like "kg", "tons" or equivalent.

Thank you for your suggestion, as suggested also from the other reviewers we have modified this section of the manuscript to make it more consistent with the scope of this study.

2) Page 5, photocatalyst characterization in section 2.2: For the XPS analysis, some more details on the experimental conditions and the data analysis are needed: What is the actual energy resolution of the instrument? What kind of background subtraction was used for quantification (i.e. linear, Shirley, Tougaard)? What are the photoionization cross sections used for quantification? Which software was used? Please add related information!

Thank you for the comment. We have added information in the manuscript and modified the XPS data presentation.

3) Page 6, section 2.3, line 128: Please include "... and the to a photomultiplier as detector".

Thank you for your comment. We have included your suggestion to the experimental section.

4) Line 136/137: "... and the data points were reduced to 200." This is unclear to me - please specify.

Thank you for your comment. This sentence means that the number of saved data points is reduced to 200 based on the used software. In general, this is an automatically step, where raw data points are reduced to less points in the saved file.

5) Page 7, XRD analysis: How were the lattice parameters determined? Did the authors used a Rietveld refinement or an entire pattern fitting model? Which software was used, and what are the uncertainties of the determined lattice parameters - please add information.

Thank you for your comment. The baseline was adjusted and the position of some peaks. No refinement has been done in the patterns and the software used to present the results is Diffrac.EVA. About the reference, we can supply the following data: Calculated from ICSD using POWD-12++ (2004), Chabot, B., Parthe, E., Acta Crystallogr., Sec. B, volume 34, page 645 (1978).

6) Page 8, Fig. 2: Please adapt the scaling of the patterns from the three diffractograms to the same height, the pattern of the most interesting actual sample is the smallest.

Thank you for the suggestions, we modified the presentation of the data to make it more clearly for readers. For further information, we are referring to the previous work in the manuscript.

7) Page 8, line 174-176: What are the uncertainties for the quantification? Please specify an error estimate.

Thank you for the suggestions, we have added the information to the manuscript.

8) Page 9, Fig. 3: The scaling and the labelling of the EDX are far too small, please enlarge labels.

Thank you for your comment. We agree with your point and we have modified the figure.

10) Page 9, line 194: Remove "photoemissive" and insert "photoemission"

Thank you for your suggestion. We have changed the word in the manuscript.

9) Page 9, discussion of XPS peak energies: Please be more precise in the binding energies, at least it should read e.g. "... the Bi 4f peaks at 160.2 eV and 165.5 eV." (line 188), at least this is what the figure suggests to me.

11) Page 10, line 196-198: What are the errors of the quantitative XPS analysis? I cannot believe those values, as the background lines in Fig. 4 are highly erroneous in particular for the I 3d but especially for the Cs 3d lines, the background cuts substantial parts of the peaks. Furthermore, are the entire peak groups used for the quantification, i.e. both peaks for Bi3+ and Bi metal?

12) Page 10, Fig. 4(a), please label the detected and assigned peaks. Please also justify the use of three peaks for the deconvolution of the I 3d. Where the plasmon peaks for Cs included in the quantification? The Figure caption is erroneous as (C) and (D) are mixed up.

Thank you very much for the careful attention on the details of the data. We simplified the presentation of the mentioned results. We referred to the previous work for further characterization. We kindly thank you for the efforts you have made to explain your comments.

13) Page 15/16, the numbering of the equations is wrong, eq. 1 was already used on page 6, and thus it should read eq. 2, 3 and 4 on the latter pages.

Thank you for your suggestion. Based on the comments of the others reviewers we have decided to remove the mechanism part form the paper, since more experiments for clearly describing the steps are necessary, which will be a part of future works.

14) There are several typos and misspellings:

page 1, line 16 "structuralpropertiesof"  -- please use three words!

page 2, line 35 "Contaminants"

page 2, line 38 "were" instead of "was"

page 2, line 43 "strongly" (the "g" is missing)

page 3, line 73 it should read "recovered"

page 3 last line, page 4, first line: it should read " ..., its alternating ..."?

page 7, line 155: please remove one point (..)

page 8, line 171, it should read "crystallized"

page 8, line 172, it should read "... and thus increase ..."

page 9, line 181, it should read "sample holder" instead of just "holder"

page 9, line 187, it should read "probably"

page 9, line 192, it should read "species"

page 12, line 240, it should read "Contrarily"

page 12, line 241, it should read "...which is an indication for the presence of electrons ..."

page 13, line 247, it should read "usually"

page 14, line 267, it should read "in the presence of"

page 15, line 286, it should read "three processes"

Thank you very much for the careful suggestions given. We have modified all concerns in the manuscript.

Provided that the authors seriously consider the comments given above and provide a revised mansucript accordingly, I think that their work can be a nice contribution to "nanomaterials" - although their particles are in fact not in the "nano-" but in the "microscale"

Reviewer 2 Report

The authors adequately corrected the main body text of the manuscript according to the reviewer's remarks.

Author Response

(The authors gave the same response as above.)
